# Association between Lysosomal Dysfunction and Obesity-Related Pathology: A Key Knowledge to Prevent Metabolic Syndrome

**DOI:** 10.3390/ijms20153688

**Published:** 2019-07-27

**Authors:** Yuhei Mizunoe, Masaki Kobayashi, Ryoma Tagawa, Yoshimi Nakagawa, Hitoshi Shimano, Yoshikazu Higami

**Affiliations:** 1Department of Internal Medicine (Endocrinology and Metabolism), Faculty of Medicine, University of Tsukuba, Tsukuba, Ibaraki 305-8575, Japan; 2Laboratory of Molecular Pathology and Metabolic Disease, Faculty of Pharmaceutical Sciences, Tokyo University of Science, 2641 Yamazaki, Noda, Chiba 278-8510, Japan; 3International Institute for Integrative Sleep Medicine (WPI-IIIS), University of Tsukuba, Tsukuba, Ibaraki 305-8575, Japan; 4Life Science Center, Tsukuba Advanced Research Alliance (TARA), University of Tsukuba, Tsukuba, Ibaraki 305-8577, Japan; 5Japan Agency for Medical Research and Development–Core Research for Evolutional Science and Technology (AMED-CREST), Chiyoda-ku, Tokyo 100-1004, Japan

**Keywords:** lysosome, cathepsin, adipose tissue, liver, lysosomal dysfunction

## Abstract

Obesity causes various health problems, such as type 2 diabetes, non-alcoholic fatty liver disease, and cardio- and cerebrovascular diseases. Metabolic organs, particularly white adipose tissue (WAT) and liver, are deeply involved in obesity. WAT contains many adipocytes with energy storage capacity and secretes adipokines depending on the obesity state, while liver plays pivotal roles in glucose and lipid metabolism. This review outlines and underscores the relationship between obesity and lysosomal functions, including lysosome biogenesis, maturation and activity of lysosomal proteases in WAT and liver. It has been revealed that obesity-induced abnormalities of lysosomal proteases contribute to inflammation and cellular senescence in adipocytes. Previous reports have demonstrated obesity-induced ectopic lipid accumulation in liver is associated with abnormality of lysosomal proteases as well as other lysosomal enzymes. These studies demonstrate that lysosomal dysfunction in WAT and liver underlies part of the obesity-related pathology, raising the possibility that strategies to modulate lysosomal function may be effective in preventing or treating the metabolic syndrome.

## 1. Introduction

Obesity increases disturbances in the metabolic, endocrine and immune systems in the body due to over-nutrition, resulting in reduced life expectancy and/or increased health problems including type 2 diabetes, non-alcoholic fatty liver disease (NAFLD), hyperlipidemia, hypertension, and cardio- and cerebrovascular diseases [1,2,3]. Obesity induces numerous cellular stresses and inflammatory signaling pathways by ectopic accumulation of fat in various tissues, leading to insulin resistance, pancreatic β-cell dysfunction, and hepatic steatosis [3,4]. Among the metabolic organs, white adipose tissue (WAT) and liver are especially implicated in energy imbalance and obesity-related pathology.

WAT consists of several cell types: adipocytes, mesenchymal stem cells, fibroblasts, macrophages, and other immune cells [5]. Adipocytes account for the majority of WAT and store excessive energy in the form of triglyceride (TG). WAT also functions as an endocrine organ that secretes ADIPOQ/adiponectin, leptin, and pro-inflammatory cytokines such as interleukin-6 (IL-6), tumor necrosis factor α (TNFα), serpin family E member 1 (SERPINE1/PAI1) and CC motif chemokine ligand 2 (CCL2)/monocyte chemoattractant protein-1 (MCP1) [5,6]. ADIPOQ activates AMP-activated protein kinase (AMPK) in skeletal muscle and liver to enhance fatty acid metabolism and glucose uptake, thereby improving insulin resistance [5,7]. In contrast, TNFα, IL-6, CCL2 and SERPIN1 are associated with inflammatory reactions and insulin resistance and are positively correlated with obesity [8,9,10,11]. The characteristics of WAT and its secretory profile vary depending on the size of adipocytes. Hypertrophic adipocytes, which store a large amount of TG and are frequently observed in obese individuals, secrete pro-inflammatory adipokines rather than anti-inflammatory adipokines like ADIPOQ and recruit immune cells with pro-inflammatory function [3,5]. These infiltrating immune cells then secrete more pro-inflammatory cytokines. Therefore, in obese WAT, the interaction between hypertrophic adipocytes and infiltrating inflammatory cells becomes a vicious cycle, leading to persistent and continuous low-grade inflammation.

The liver is the central metabolic organ that regulates key aspects of glucose and lipid metabolism, including gluconeogenesis, fatty acid β-oxidation, lipoprotein uptake and secretion, and lipogenesis [12,13]. NAFLD is a representative hepatic metabolic disease characterized by accumulation of TG and free fatty acids (FFAs) in hepatocytes. NAFLD is classified into simple fatty liver and more advanced non-alcoholic steatohepatitis (NASH), which develops into cirrhosis and hepatocellular carcinoma with inflammation and fibrosis [14]. The prevalence of NAFLD in developed countries is estimated at approximately 20% to 30%, of which 2% to 3% is NASH [15,16,17]. NAFLD/NASH is not a single disease but instead a syndrome that encompasses various pathological conditions. Despite ongoing research on NASH/NAFLD, the detailed onset mechanisms and treatment strategies for NASH/NAFLD remain unclear. One of the accepted onset mechanisms of NAFLD/NASH is the “second-hit hypothesis” [18]. First, fatty liver progresses due to obesity (first hit). Second, inflammation and fibrosis are triggered by various hepatocellular injury factors such as oxidative stress and influx of FFAs from adipocytes (second hit), resulting in the progression to NASH. Recent studies have suggested that more steps initiated from fatty liver are involved in the pathogenesis of NASH/NAFLD and this has been called the “multiple hits hypothesis” [19,20]. Hypertrophic adipocytes have been shown to leak lipids, which induces ectopic lipid accumulation in liver. Thus, a better understanding of the close link between the characteristic changes in WAT and liver is important for obesity-related pathology.

## 2. Overview of Lysosomes and Lysosome-Associated Diseases

Lysosomes are intracellular organelles common to eukaryotes that contain many hydrolytic enzymes such as nucleases, glycosidases, lipases, and more than 20 types of proteases [21,22]. Since the discovery of lysosomes by Christian de Duve in 1950, lysosomes have been shown to play an important role in regulating and maintaining cell function, including via functions in endocytosis, exocytosis, and autophagy [21,22,23]. Lysosomes are limited by a single phospholipid layer (7–10 nm) and have an average diameter of 0.5–1.0 µm [24]. The lumen is maintained at approximately pH 4.5–5.0, which is established by the vacuolar H^+^-ATPase (v-ATPase)—an ATP-driven proton pump.

Lysosomal proteins are divided into two groups: lysosomal membrane proteins and hydrolases. Over 100 lysosomal membrane proteins have been described, many of which are highly glycosylated toward the lumen to escape from degradation by hydrolytic enzymes. The roles of these lysosomal membrane proteins involve the maintenance of acidic conditions in lysosomes; the transport of metabolites, ions and hydrolases across the membrane; and the regulation of membrane fusion events [25]. The lysosomal lumen contains approximately 60 hydrolases that can break down biopolymers/biomolecules (proteins, lipids, carbohydrates, etc.) into constituent units (amino acids, phospholipids, sugars, and nucleic acids).

Recent reports showed that lysosomes function as a metabolic signaling hub by interacting with the mechanistic target of rapamycin complex 1 (mTORC1) and transcription factors, such as transcription factor EB (TFEB) [22,26]. TFEB is a member of the microphthalmia family (MiT family) of basic helix-loop-helix leucine-zipper (b HLH-Zip) transcription factors and a master regulator of lysosomal biogenesis and autophagy [27,28]. TFEB is normally phosphorylated in the cytoplasm by the mTORC1 complex on lysosomal membranes. However, under starvation and lysosomal stress conditions, TFEB is dephosphorylated and translocated into the nucleus, thereby inducing gene expression. In the nucleus, TFEB directly binds to promoter regions that contain a coordinated lysosomal expression and regulation (CLEAR) element, a common 10 base pair E-box-like palindromic sequence and upregulates lysosome-related genes [29]. Indeed, TFEB overexpressing cells display an increase in the expression of genes encoding lysosomal enzymes and membrane proteins and a number of lysosomes [30]. Of note, TFEB exerts global transcriptional control on lipid catabolism via PPARG coactivator 1 alpha (Ppargc1α)/PGC1α and peroxisome proliferator activated receptor alpha (P parα) during starvation [31], indicating that a lysosomal regulator serves as a metabolic sensor. These findings indicate the importance of investigating the involvement of lysosomal dysfunction in metabolic organs.

Autophagy is a cellular process closely related to lysosomal function. Autophagy is an intracellular lysosome-mediated degradation system that contributes to cell survival, maintenance, and differentiation. Autophagy is generally classified into three types; macroautophagy, chaperone-mediated autophagy (CMA), and microautophagy [32,33,34]. However, in this text, the term “autophagy” refers to macroautophagy. In autophagy, intracellular proteins and organelles are surrounded by the isolation membrane and fused to lysosomes to form autolysosomes. The contents in autolysosomes are then degraded by hydrolytic enzymes in lysosomes [35,36].

Previous analyses of several autophagy-deficient mice have revealed that autophagy abnormalities are involved in the onset of various diseases such as cancer, neurodegenerative diseases and metabolic diseases [35,37]. At present, many autophagy-related diseases with polymorphisms/mutations in autophagy genes in humans have been identified [38]. For example, mutation of the *WDR45* gene has been reported as a cause of static encephalopathy of childhood with neurodegeneration in adulthood (SENDA), a neurodegenerative disease with iron deposits in the brain. The *WDR45* gene encodes the WIPI4 protein (a human homolog of yeast Atg18), which is essential for autophagosome formation [39,40]. Other autophagy-related diseases are suspected to be associated with disturbed autophagosome formation and these are extensively discussed elsewhere [35,36].

Lysosomal storage disease is a well-known group of autophagy-related diseases. Lysosomal storage disease results from genetic mutation of lysosomal enzymes. In lysosomes with deleted or inactivated enzymes, transported substrates cannot be appropriately degraded and instead abnormally accumulate, leading to various diseases [41,42]. For example, Pompe disease, a metabolic myopathy characterized by the deficiency of α-glucosidase, accumulates glycogen in lysosomes of cardiac muscle, skeletal muscle and liver [43]. Moreover, Niemann-Pick disease type C is a neurodegenerative disorder with mutation in the *NPC1* or *NPC2* (Niemann-Pick disease, type C1 or C2) genes, which encode proteins involved in cholesterol transport. Mutation in *NPC* genes causes abnormal cholesterol accumulation in lysosomes [44]. Despite differences in the degree of disease progression or the time of onset, lysosomal diseases are often associated with severe symptoms.

MVB (multivesicular body) is a form of late endosomes, the contents of which are then transported into lysosomes [45]. Interestingly, Zhao et al. highlighted that the MVB-lysosomal pathway contributes to steatohepatitis through lysosomal degradation of Toll-like receptor 4 (TLR4), which is previously reported to be important for the progression of NASH [46,47]. The authors revealed that transmembrane BAX inhibitor motif-containing 1 (TMBIM1) facilitates MVB formation and promotes the lysosomal degradation of TLR4, suppressing the inflammation in liver [46]. As is evident from this report, lysosome-related pathway can play a key role in metabolic diseases. Additionally, recent studies have reported that lysosomal dysfunction occurs in metabolic disorders, indicating the need to better understand lysosomal dysfunction in metabolic organs.

## 3. Lysosomal Proteases

The lysosomal proteases include cathepsin, legumain, napsin, and tripeptidyl-peptidase I (TPP1) [48,49,50]. Cathepsin is a representative lysosomal protease family with many members, from cathepsin A to cathepsin Z [51,52,53]. While cathepsin B, H, L, C, X, F, O and V are ubiquitously expressed in almost all tissues, the expressions of cathepsin K (osteoclast) and cathepsin W (CD8+ lymphocytes and natural killer (NK) cells) are limited to specific cells or tissues [53]. Cathepsins are generally synthesized as pro-cathepsins, modified in the endoplasmic reticulum and Golgi apparatus and then transported to lysosomes. In the acidic environment in lysosomes, pro-cathepsins are matured into the active form either by cleavage by other enzymes or auto-modification by its own protease activity (cathepsin maturation) [54].

Cathepsins are roughly classified into three types according to the type of amino acid in the active center: serine proteases (cathepsin A and G), aspartic proteases (cathepsin D (CTSD) and E) and cysteine proteases (cathepsin B (CTSB), L (CTSL) and many other cathepsins) [55,56]. CTSB, CTSL and CTSD are the most abundant cathepsins in tissues [57]. CTSL only has endopeptidase activity, whereas CTSB has endopeptidase and carboxydipeptidase activity [52,53,58]. Similar to other cathepsins, CTSL is synthesized as an inactive form (39 kDa), transported to endosomes and lysosomes and converted into the active form (25–30 kDa) [54,59,60,61]. CTSB is also synthesized as an inactive form (44 kDa) but is converted into a 33 kDa single-chain form at the transport step. In lysosomes, this form is matured into the active two-chain form composed of 24–27 kDa and 5 kDa polypeptides [54,62,63,64,65]. CTSB can also exhibit stable enzyme activity even under more neutral conditions [66].

Previous studies using animal models demonstrated that CTSB-deficient mice show no obvious phenotype, whereas CTSL-deficient mice display periodic hair loss and bone developmental disorder and develop a progressive dilated cardiomyopathy in the heart [67,68,69]. Petermann et al. reported that CTSL deficiency affects lysosomal function by increasing the number of lysosomes and changing the morphology of lysosomes in the mouse heart, despite the absence of lysosomal storage accumulation [69]. Moreover, cardiomyocyte-specific exogenous expression of CTSL in CTSL-deficient mice resulted in improved cardiac contraction, normal heart weight, and regular ultrastructure of cardiomyocytes [70]. These reports support the necessity of CTSL for lysosomal function. CTSL and CTSB double-deficient mice exhibit postnatal cerebral atrophy and die at 2 to 4 weeks of age [71], indicating that the mutual interaction of cathepsins in lysosomes maintains lysosomal function. In addition to the loss of function studies described above, abnormal activation of cathepsins has been associated with various pathological conditions, such as cancer [55], kidney disease [72], neurodegenerative disease [73], and autoimmune diseases [74]. Notably, recent several studies have addressed the influence of cathepsin abnormalities on obesity-related pathology in metabolic organs such as WAT and liver (Table 1). Therefore, we focused on cathepsin abnormalities, especially CTSL, CTSB and CTSD, as a mechanism of obesity-related lysosomal dysfunction in WAT and liver.

## 4. Lysosomal Dysfunction in Obese Adipose Tissue

In this section, we outline the significance of lysosomal dysregulation in obese WAT mainly based on our recent report. We recently demonstrated that in WAT of either high-fat diet (HFD)-induced or genetically obese (*ob/ob*) mice, the levels of active CTSL (25–30 kDa) and its enzyme activity were significantly decreased compared with control mice, while inactive CTSL (50 kDa) and its mRNA expression levels were significantly increased [78]. These results indicated that obesity causes a decline in CTSL maturation, resulting in the downregulation of CTSL activity. We also observed accumulation of autophagosomes in both obese WAT and CTSL-knockdown 3T3L1 adipocytes [78], indicating that lysosomal dysfunction due to decreased CTSL activity may contribute to autophagic abnormality in obese WAT. These data are consistent with previous studies that showed that autophagy flux was decreased in WAT of HFD mice and hypertrophied 3T3L1 adipocytes, resulting in induced expression of inflammatory cytokines such as IL1B, IL6, and CCL2 [83,84]. Furthermore, Soussi et al. identified death-associated protein kinase 2 (DAPK2) as a regulatory factor of lysosomes [77]. DAPK2 is expressed mainly in mature adipocytes rather than stromal vascular cells in WAT, and DAPK2 mRNA levels are strongly downregulated according to obesity status in human and mice and gradually recover after bariatric surgery-induced weight loss [77]. The authors showed that DAPK2 modulates lysosome remodeling and constitutive autophagy in adipocytes. Hence, lysosomal dysfunction in adipocytes, followed by the destruction of autophagic clearance, may contribute to pathology in both obese WAT in mouse and human.

Another study showed that expressions of CTSL and CTSB were increased in the fat interstitial vascular cell fraction of epididymal WAT of obese Zucker rats instead of the mature adipocyte fraction [76]. The authors also showed that rosiglitazone, a peroxisome proliferator activated receptor gamma (PPARG)/PPARγ agonist that accelerates adipocyte differentiation, attenuated the expression of precursor and mature forms of CTSL in the epididymal fat depot, but failed to alter the expression of CTSB [76]. Thus, alterations of CTSL and CTSB expressions in WAT are very complicated, likely depending on the degree of obesity status, type of food, timing of tissue sampling, or animal species.

The maturation of cathepsins is generally regulated by lysosomal pH, other proteases, and endogenous cathepsin inhibitors such as cystatins, thyropins, and serpins, which are associated with obesity [53]. For example, cystatin C, an endogenous cathepsin inhibitor secreted from WAT, is significantly increased in serum of obese individuals [85]. Interestingly, p41 invariant chain (splicing variant of CD74) of MHC class II selectively suppresses CTSL as an endogenous inhibitor [53,86]. This finding indicated that the p41 invariant chain may be associated with obesity-induced downregulation of CTSL in WAT despite no direct evidence at present. In summary, although the exact causes and mechanisms are still unclear, lysosomal dysfunction represented by downregulated CTSL in WAT or adipocytes abrogates autophagy and can participate in obesity-related pathology.

In our previous study, we also identified the complementary activation of CTSB induced by the downregulation of CTSL in obese WAT [78]. Activated CTSB was shown to induce the inflammasome, which was also observed in 3T3L1 adipocytes overexpressing CTSB. In agreement with our results, Gornicka et al. reported that HFD-feeding significantly enhanced the activity of CTSB and evoked the inflammatory response and apoptosis in obese WAT [75]. In parallel, the authors demonstrated that CTSB-deficient mice exhibited reduced lysosomal abnormality and cell death of adipocytes and macrophage infiltration to WAT [75]. These data indicate the relationship between CTSB and inflammatory response in obese WAT. In addition, Ju et al. reported that CTSB protein was significantly increased in adipocytes of epididymal WAT during the immune response [79].

The inflammasome is a protein complex composed of pattern recognition receptors such as NLR family pyrin domain containing 3/NLRP3, PYCARD/ASC (PYD and CARD domain containing), and caspase 1/CASP1. The inflammation signal via the inflammasome is as follows. Initially, the NLRP3 receptor is activated by various stimuli and forms a complex of inflammasome with pro-CASP1 via the adaptor protein PYCARD. The formation of the inflammasome allows cleavage of pro-CASP1 via an autolytic action and its maturation into active CASP1. Finally, active CASP1 cleaves and maturates the precursor forms of IL1B and IL18 into mature forms, which are pro-inflammatory cytokines to induce inflammatory reaction and macrophage infiltration [87]. In obese WAT, the NLRP3 receptor is stimulated by metabolites (such as urate crystals, cholesterol crystals, FFAs, etc.), followed by activation of the inflammasome complex [88,89]. Interestingly, a mechanism of inflammasome reaction involving CTSB was also reported. CTSB is released through lysosomal membrane permeabilization (LMP), which is caused by FFA or ceramides, and activates CASP1 by interacting with the inflammasome complex [90,91]. The release of CTSB by LMP is a phenomenon observed during induction of apoptosis [92]. Therefore, the influence of CTSB activation on the inflammasome reaction may be associated with a crown-like structure characterized by infiltrated macrophages and apoptotic adipocytes in obese WAT.

In addition to inflammasome activation, we found a relationship between CTSB overexpression and decreased protein levels of perilipin 1 (PLIN1), a protein that coats lipid droplets in adipocytes. PLIN1 downregulation is suspected to enhance the release of FFAs, probably resulting in migration of macrophages to WAT (manuscript in preparation). Taken together, CTSL downregulation leads to complementary CTSB activation that can contribute to obesity-induced inflammation reactions in WAT (Figure 1). Thus, downregulation of a lysosomal enzyme is likely to evoke abnormal activation of another enzyme, which contributes to lysosomal dysfunction.

## 5. Lysosomal Dysfunction in Obese Liver

In recent years, many reports have suggested the involvement of lysosomal dysfunction and obesity in the liver. Chronic inflammation and lysosomal dysfunction are known to coexist in the obese liver. The protein expression levels and activity of CTSB and CTSL are reduced in primary culture hepatocytes from genetically obese mice [80]. Inami et al. reported that the activity of both CTSB and CTSL in autolysosomes was suppressed in *ob/ob* mice because of the decreased lysosomal acidification [80]. Recently, Wang et al. reported increased expression of asparagine synthetase (ASNS) due to endoplasmic reticulum stress and subsequent lysosomal calcium retention affected the lysosomal acidification [82]. The authors also reported that steatohepatitis resulted in accumulation of the precursor form of CTSD, whereas the reduced mature form of CTSD was detected in both C57BL/6 wild and *db/db* mice [82]. In addition, another study showed that CTSB, CTSD and CTSL expressions were significantly suppressed in the liver from NAFLD patients, which correlated with hepatic inflammation [81]. In agreement with these studies, it was uncovered that plasma CTSD levels were reduced in children with NASH [93]. Conversely, Walenbergh et al. documented that adult patients with NASH exhibited an increase of plasma CTSD levels compared to adults without hepatic inflammation [94]. Furthermore, the authors underscored preventive effects of a CTSD inhibitor on obesity-induced hepatic inflammation and disturbance in lipid metabolism in mice [95]. These findings indicate that lysosomal dysfunction accompanied by suppression of expression or maturation of cathepsin is involved in the onset of hepatic metabolic disorder and NAFLD.

Lipotoxicity is a condition in which excess lipid accumulation in non-adipose tissue disrupts cell function and increases cell death [96,97]. Failure of packaging excess lipid into lipid droplets causes chronic elevation of circulating fatty acids, which can reach toxic levels within non-adipose tissues [96,97]. Such accumulated ectopic lipid provokes insulin resistance in skeletal muscle and liver, leading to type 2 diabetes, fatty liver and heart disease [98,99]. In particular, saturated fatty acid-induced lipotoxicity has been proven to be a mechanism in lysosomal dysfunction [100,101]. Saturated fatty acid induced LMP, followed by BAX (BCL2 associated X, apoptosis regulator)-associated cell death. LMP is a key mechanism by which chronic lipid overload promotes lysosome dysfunction and triggers apoptotic cell death. Moreover, the fusion of autophagosomes with lysosomes was impaired in an endoplasmic reticulum stress-dependent manner in cultured hepatocytes after saturated FFA treatment, resulting in suppression of degradation of autophagosomes [102]. A potential mechanism for this dysfunction may involve increases in intracellular lipids that alter the lipid composition of intracellular membrane on both autophagosomes and lysosomes [103]. These findings indicate that obesity-induced lipotoxity disrupts the hepatic lysosomal function and can trigger the onset or progression of NAFLD.

Lysosomal lipase (LIPA/LAL) deficiency (LAL-D) is another example of hepatic lysosomal dysfunction associated with obesity. LAL-D is a genetic, chronic, and progressive metabolic disease that is characterized by multiorgan damage and premature death due to uncontrolled accumulation of cholesteryl esters (CEs) and TGs [104]. LAL is the lysosomal enzyme primarily responsible for the hydrolysis of CEs and TGs in lipoproteins into free cholesterol (FC) and FFAs. LAL-D also includes Wolman disease (WD) and CE storage disease (CESD) [105]. LAL-D patients exhibit pathological phenotypes in liver, such as elevated ALT, enlarged liver fibrosis and/or cirrhosis, which account for 50% of the reported liver-related deaths in patients younger than 21 years of age [106,107]. Du et al. reported that LAL-deficient mice display progressive hepatosplenomegaly and massive TG accumulation in liver [108]. In contrast, hepatocyte-specific LAL-deficient mice (Liv-Lipa−/−) show resistance to diet-induced obesity despite a marked increase of CE concentrations [109]. Moreover, Cahova et al. demonstrated that short-term HFD treatment induced activity of LAL and then increased production of diacylglycerol, resulting in the rapid onset of hepatic insulin resistance [110]. The results from these animal studies in determining how LAL-D participates in obese hepatic pathology are conflicting, but hepatic LAL plays a fundamental role in preventing liver damage and maintaining lipid and energy homeostasis. However, a recent study showed that hepatic LAL activity was decreased in a cohort of adult NAFLD patients with more severe symptoms than in the remaining NAFLD population [111,112]. Taken together, these findings suggest that LAL-D is at least associated with NAFLD progression.

Nitric oxide 2 (NOS2), called “iNOS,” is also involved in obesity-related lysosomal dysfunction. Qian et al. reported that lysosomal NOS2-mediated NO signaling disrupts hepatic lysosomal function, contributing to obesity-associated defective hepatic autophagy and insulin resistance [113]. The authors also reported that obesity promotes NOS2 localization to lysosomes and results in accumulated lysosomal NO in the liver and that the overproduction of lysosomal NO exacerbates lysosomal nitrosative stress with impairment of lysosomal function and autophagy. Collectively, these studies indicate that abnormality of lysosomal enzymes other than proteases can also play an important role in NAFLD pathology (Figure 2).

## 6. Conclusions

In this review, we highlighted the association between lysosomal function and obesity in WAT and liver based on our study and those of others. In obese WAT and liver, lysosomal dysfunction systemically causes blocked autophagy, chronic inflammation and insulin resistance. As described above, most studies on lysosomal dysfunction in obese WAT focus on autophagy or cathepsins. However, in liver, downregulation of enzymes other than cathepsins are also reported to be associated with obesity. Together these studies indicate that lysosomal dysfunction in WAT and liver will be a therapeutic target in obesity-related pathology and suggest that therapeutic strategies designed to modulate lysosomal function may be beneficial in the prevention or treatment of the metabolic syndrome.

## Figures and Tables

**Figure 1 ijms-20-03688-f001:**
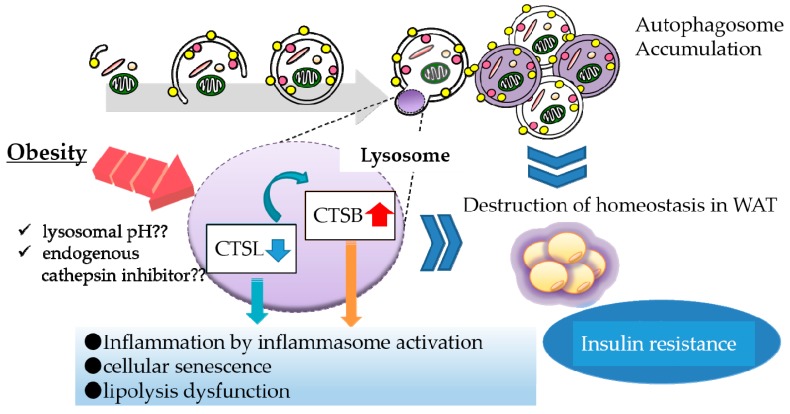
Lysosomal dysfunction in obese adipose tissue. In obese adipose tissue, various stresses such as oxidative stress and lysosomal pH abnormality attenuate the maturation of cathepsin L (CTSL) protein, leading to complementary activation of CTSB. Downregulation of CTSL protein causes the accumulation of autophagosomes, resulting in suppression of autophagic clearance. Upregulation of CTSB protein enhances inflammation by activating the inflammasome complex. These lysosomal alterations consequently affect the function of white adipose tissue (WAT).

**Figure 2 ijms-20-03688-f002:**
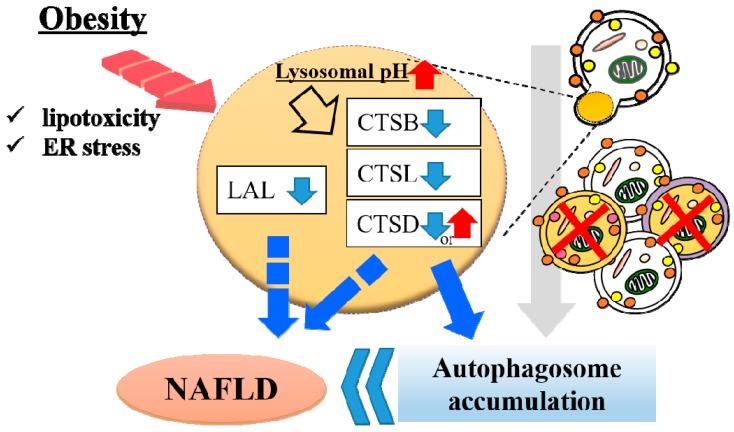
Lysosomal dysfunction in obese liver. In obese liver, various stresses such as lipotoxicity and endoplasmic reticulum (ER) stress alkalinizes the lysosomal pH, resulting in downregulated activity of cathepsin L (CTSL), CTSB and CTSD. Downregulation of cathepsins induce autophagosome accumulation due to impaired lysosomal clearance. Hepatic lysosomal lipase (LAL) is associated with the progression of non-alcoholic fatty liver disease (NAFLD). These lysosomal abnormalities contribute to the onset of NAFLD.

**Table 1 ijms-20-03688-t001:** Involvement of lysosomal dysfunction in metabolic tissues.

Metabolic Organ	Characteristics	Results	Reference
Adipose tissue (WAT)	Upregulation of CTSB	Increased lysosomal permeablization in adipocytes and contribution to cell death	Gornicka et al., 2012 [75]
	Upregulation of CTSB	Involvement in mediating the inflammatory response in cholesterol trafficking	Hannaford et al., 2013 [76]
	Downregulation of DAPK2	Modulation of Lysosome-Mediated Remodeling	Soussi et al., 2015 [77]
	Downregulation of CTSL	Autophagosome accumulation	Mizunoe et al., 2017 [78]
	Complementary upregulation of CTSB	Inflammasome activation in obese WAT	
	Upregulation of CTSB	Contribution to the pathogenesis of obesity-related inflammation	Ju et al., 2019 [79]
Liver	Downregulation of CTSB	Autophagosome accumulation in liver from *ob/ob* mice	Inami et al., 2011 [80]
	Downregulation of CTSL		
	Downregulation of CTSL, CTSB	patients with NAFLD	Fukuo et al., 2014 [81]
	Reduced lysosomal acidity Dysfunction of CTSD maturation	Defective lysosomal clearance of autophagosomes	Wang et al., 2018 [82]

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
