# Peer review of "Association between Lysosomal Dysfunction and Obesity-Related Pathology: A Key Knowledge to Prevent Metabolic Syndrome"

_ijms, 2019, doi:10.3390/ijms20153688_

Reviewer 1 Report

Based upon authors' studies on hepatic lysosome study, they reviewed the importance of lysosomal dysfunction in obese-induced NASH/NAFLD.  The review is well written, however, the followings should be mentioned to make this review comprehensive.

1. Toll-like receptor 4 (TLR4) is also plays a key role in multivesicular body (MVB)-lysosomal pathway and NASH development (Gut 61, 1058–1067 (2012)). Recently it is reported that transmembrane BAX inhibitor motif-containing 1 (TMBIM1) is an effective suppressor of steatohepatitis and MVB-lysosomal pathway(Nature Medicine 23, 742–752 (2017)).  This pathway should be mentioned in section of Overview of lysosomes and lysosome-associated disease.

2. The role of cathepsin D in NASH is still controversial and authors should refer properly ie Sci Rep. 2017 Jun 14;7(1):3494. doi: 10.1038/s41598-017-03796-5. In this article, they claim that blocking of cathepsin D is effective in prevention of fibrosis. The authors should mention on the role of cathepsin D in LL. 264-270.

Author Response

Reviewer #1: Based upon authors' studies on hepatic lysosome study, they reviewed the importance of lysosomal dysfunction in obese-induced NASH/NAFLD. The review is well written, however, the followings should be mentioned to make this review comprehensive.

1. Toll-like receptor 4 (TLR4) is also plays a key role in multivesicular body (MVB)-lysosomal pathway and NASH development (Gut 61, 1058–1067 (2012)). Recently it is reported that transmembrane BAX inhibitor motif-containing 1 (TMBIM1) is an effective suppressor of steatohepatitis and MVB-lysosomal pathway (Nature Medicine 23, 742–752 (2017)). This pathway should be mentioned in section of Overview of lysosomes and lysosome-associated disease.

2. The role of cathepsin D in NASH is still controversial and authors should refer properly ie Sci Rep. 2017 Jun 14;7(1):3494. doi: 10.1038/s41598-017-03796-5. In this article, they claim that blocking of cathepsin D is effective in prevention of fibrosis. The authors should mention on the role of cathepsin D in LL. 264-270.

Our response:

We appreciate the reviewer #1’s helpful suggestion and introduction of very interesting and suggestive articles.

To address the reviewer’s suggestion, we described the involvement between MVB-lysosomal pathways and steatohepatitis in the section of “Overview of lysosomes and lysosome-associated disease” and controversial issues in the role of cathepsin D in NASH in the section of “Lysosomal dysfunction in obese liver”. Moreover, we added the related references. The locations of these corrections are as follows.

p3; lines 138–146.

p7; lines 276–283.

p11; lines 461–469.

p13; lines 596–607.

Reviewer 2 Report

The manuscript entitled "Functional dysregulation of lysosomes in metabolic organs in obesity-related pathology" is a good review about thie association between lysosomal function and obesity. It is very interesting.

In my opinion the title is not very appropiate I am suggesting an alternative title more attractive for the lectors as

Association between lysosomal dysfunction and obesity related pathology. A key knowledge to prevent metabolic syndrome.

or another one...

Author Response

Reviewer #2: The manuscript entitled "Functional dysregulation of lysosomes in metabolic organs in obesity-related pathology" is a good review about the association between lysosomal function and obesity. It is very interesting.

In my opinion the title is not very appropiate I am suggesting an alternative title more attractive for the lectors as

Association between lysosomal dysfunction and obesity related pathology. A key knowledge to prevent metabolic syndrome.

or another one...

Our response:

We appreciate the reviewer’s helpful suggestion. The suggested title “Association between lysosomal dysfunction and obesity-related pathology: A key knowledge to prevent metabolic syndrome” is definitely more appropriate and attractive. Thus, we revised the title of this review according to the reviewer’s suggestion.